# Association of Coffee and Energy Drink Intake with Suicide Attempts and Suicide Ideation: A Systematic Review and Meta-Analysis

**DOI:** 10.3390/nu17111911

**Published:** 2025-06-02

**Authors:** Chen Ee Low, Nicole Shi Min Chew, Sean Loke, Jia Yang Tan, Shayne Phee, Ainsley Ryan Yan Bin Lee, Cyrus Su Hui Ho

**Affiliations:** 1Yong Loo Lin School of Medicine, National University of Singapore, Singapore 119228, Singapore; 2Department of Psychological Medicine, Yong Loo Lin School of Medicine, National University of Singapore, 1E Kent Ridge Road, NUHS Tower Block, Level 9, Singapore 119228, Singapore; 3Department of Psychological Medicine, National University Hospital, Singapore 119228, Singapore

**Keywords:** caffeine, coffee, energy drinks, suicide, suicidal ideation

## Abstract

Introduction: Caffeine, in the form of coffee, tea and energy drinks, is recognised as the world’s most utilised psychoactive substance and consumed by approximately 80% of the global population daily. Emerging studies have suggested a more complex relationship in terms of the mental health outcomes that can arise after consumption. This is the first systematic review and meta-analysis that aims to explore the effects of caffeine consumption on the risk of suicide attempts, ideation and self-harm. Methods: This PRISMA-adherent systematic review involved a systematic search of PubMed, Embase, Cochrane and PsycINFO for all studies that evaluated the effects of caffeine consumption on the risk of suicide attempts, ideation and self-harm. Random effects meta-analyses and meta-regression were used for primary analysis. Results: Seventeen studies were included. The results demonstrated that coffee consumption of more than 60 cups per month significantly decreases the risk of suicide attempts. In contrast, energy drink consumption from as low as one cup per month was significantly associated with an increased risk of both suicide attempts and ideation. Meta-regression demonstrated a strong association between the dosage consumed and suicidality outcomes. Systematic review highlighted that male gender and substance usage significantly increased caffeine consumption. Conclusion: The results studied the associations between coffee and energy drink intake with suicide risk and suicidal ideation. Coffee intake was associated with reduced odds of suicide ideation and attempts, while consuming energy drinks was associated with an increased risk of both adverse outcomes. Further studies would be essential to elucidate the psychosocial factors and causative links underlying this association. Understanding the relationship between caffeine consumption and mental health outcomes is crucial to develop public health strategies to boost the mental health of consumers.

## 1. Introduction

Caffeine, in the form of coffee, tea and energy drinks, is often recognised as the world’s most utilised psychoactive substance [1], is consumed by approximately 80% of the global population daily [2]. According to the Kantar Worldpanel Beverage Consumption Panel survey, 43% of children two to five years of age and 100% of adults over the age of 65 consume caffeine daily, with the mean daily intake increasing steadily with increasing age [3]. While caffeine is mostly found in coffee beans, it can also be naturally found in tea leaves and cocoa beans [1] and is often added into energy drinks as a stimulant compound [4]. According to the U.S. Department of Agriculture (USDA), an 8 oz cup of brewed coffee contains approximately 95 mg of caffeine, black tea around 47 mg, green tea about 28 mg and energy drinks typically contain 80–100 mg per 8 oz, depending on the brand and formulation [5]. These products, which include coffee and energy drinks, are increasingly widespread due to heavy marketing strategies and ease of availability [6]. Although caffeine has no nutritional value [2], it is primarily consumed for its effects on memory and concentration enhancement as well as its positive impact on the consumer’s physical performance [7]. Many of its physical side effects have been thoroughly explored in numerous studies, which include an increase in nervousness, insomnia, increased urination, gastrointestinal symptoms and tachycardia [8]. Aside from the physiological effects of caffeine consumption, there may exist a bidirectional relationship between caffeine intake and adverse mental health outcomes. For example, increased caffeine intake may be observed in individuals who face increased stress in their daily lives or are physiologically dependent on caffeine [9].

The rise in incidence of suicide, suicide ideation and self-harm has been a cause of significant public health concern [10,11,12,13]. The World Health Organisation (WHO) estimates that there are 700,000 successful suicides and many more with suicidal ideations every year [14], as well as 10–20 million deliberate self-harm attempts worldwide each year [15]. Recent studies have shown that caffeine consumption is associated with worse mental health outcomes in consumers. For example, in a study by Park et al. [16], more frequent consumers of caffeinated energy drinks were found to have an increased suicide and suicide ideation risk compared to less frequent consumers. Similarly, a prospective study conducted in USA [17] found increased suicide rates amongst coffee drinkers in women. However, some studies have shown a positive outcome of caffeine intake on mental health outcomes instead. For example, Lucas et al. showed that moderate amounts of caffeine intake were found to reduce suicide risk [18]. Hence, the relationship between caffeine consumption and mental health outcomes of consumers should be more thoroughly explored.

Understanding the relationship between caffeine consumption and mental health outcomes is crucial in informing the development of public health strategies to promote awareness of the potential psychological adverse effects of caffeinated products and boost the mental health of consumers. The ubiquity of caffeine in our day-to-day consumption underscores the importance of investigating its effects on consumers’ mental health. To the best of the authors’ knowledge, this is the first systematic review and meta-analysis that aims to explore the effects of caffeine consumption on the risk of suicide attempts, ideation and self-harm.

## 2. Methods

This systematic review is reported based on the Preferred Reporting Items for Systematic Reviews and Meta-Analyses (PRISMA) guidelines. The protocol was registered on (PROSPERO: CRD42024595935).

### 2.1. Search Strategy

Literature search was performed in PubMed, Embase, Cochrane and PsycINFO. The search strategy had combined terms for ‘Caffeine’, ‘Coffee’, ‘Energy Drinks’, ‘Tea’, ‘Suicide’, ‘Suicidal Ideation’ and ‘Self-Injurious Behaviour’. Controlled vocabulary from the databases was used to identify subject headings, and synonyms with relevant truncations were applied to search for title and abstract keywords. The search strategy was adapted across different databases, with examples of the strategies used for PubMed and EMBASE found in Appendix A.

### 2.2. Inclusion and Exclusion Criteria

Two reviewers independently screened titles and abstracts of the studies for eligibility. Any discrepancies were resolved by a third reviewer. All peer-reviewed English-language studies published since inception to 8 May 2025 that evaluated the association of caffeine consumption and the risk of suicidal ideation, suicide attempts and self-harm were included. Non-empirical studies, grey literature, case reports and abstracts were excluded.

### 2.3. Data Extraction and Analysis

Two reviewers performed the extraction independently. Subject matter information included, demographics, control characteristics, amount of caffeine consumed per month and main findings of the study. The effect of caffeine on suicidality outcomes were quantified by the odds ratio (OR) or relative risk ratios (RR). The OR, RR and the 95% confidence interval (95%CI) were extracted and the standard error was calculated according to Cochrane guidelines [19]. OR was converted to RR in accordance with Cochrane guidelines [20] when more studies reporting results in RR were available for a specific suicidality outcome. The amount of caffeine consumption reported in the studies was standardised to the amount consumed per month. If a range was provided, the average number of cups was calculated.

All analyses were conducted in R (version 4.1.0) using the *meta* and *metafor* packages. A two-sided *p* value of <0.05 was considered statistically significant. Meta-analyses were performed for dichotomous outcomes to calculate either the odds or the relative risk of the suicidality outcome compared to controls. Subgroup analyses and meta-regression based on the amount of caffeine consumed per month were performed to determine if the amount of caffeine consumed monthly would influence the results.

I^2^ and τ^2^ statistics were used to assess between-study heterogeneity. An I^2^ value below 30% indicated low heterogeneity, 30% to 60% reflected moderate heterogeneity, and values above 60% represented substantial heterogeneity [21].

### 2.4. Risk of Bias

Two independent reviewers assessed the risk of bias in the included studies using the Joanna Briggs Institute (JBI) Critical Appraisal tool [22]. Any discrepancies were resolved by consulting a third reviewer. We used the GRADE framework to assess the overall certainty of evidence across studies, which includes consideration of risk of bias, inconsistency, indirectness, imprecision, and publication bias [23].

## 3. Results

From 665 records, 17 [16,17,18,24,25,26,27,28,29,30,31,32,33,34,35,36,37] studies were included (Figure 1). A total of 648 studies were excluded after removing irrelevant studies with the wrong population, study design, unrelated outcomes or duplicates. Seven studies were from Korea [16,25,28,30,31,32,35], five studies from USA [17,18,27,34,37], one study each from Germany [24], Finland [26], Turkey [29], Canada [33] and Nigeria [36], including a total of 1,574,548 participants. The mean age of the participants ranged from 15 to 48 years. Nine studies [16,28,29,30,31,32,33,34,35] investigated the effect of energy drinks, six studies [17,18,24,25,26,27] on coffee and two [36,37] on caffeine. Five studies evaluated suicidality outcomes in adolescents [16,30,31,32,35], four each on students [28,29,33,36] and the general population [18,25,26,27], one each on soldiers [34], women [17] and patients with bipolar disorder [24]. The overall characteristics of the studies can be found in Table 1.

### 3.1. Energy Drink Consumption and Suicide Attempts

Four [28,31,32,34] studies were pooled to evaluate the OR of suicide attempts after energy drink consumption. Meta-analyses of the four studies [28,31,32,34] (Figure 2) demonstrated a statistically significant increase in the risk of suicide attempts in those who consumed at least one energy drink a month compared to the comparator arm (Pooled OR = 1.81, 95%CI: 1.43–2.29). Meta-analyses of subgroup stratified by amount consumed per month revealed a linear relationship between higher frequency and higher suicide attempts. Those who consumed 21 to 30 cups per month were at highest odds of suicide attempts (Pooled OR = 2.88, 95%CI: 2.49–3.34) as compared to those who consumed 11 to 20 cups (Pooled OR = 1.61, 95%CI: 1.38–1.88) or 1 to 10 cups (Pooled OR = 1.34, 95%CI: 1.20–1.50). Meta-regression showed a significant association between a higher amount consumed per month and higher suicide attempts (Appendix A).

### 3.2. Energy Drink Consumption and Suicidal Ideation

Four studies [16,29,30,31] were pooled to evaluate the OR of suicidal ideation after energy drink consumption. Meta-analyses of the four studies [16,29,30,31] (Figure 3) demonstrated a statistically significant increase in the risk of suicidal ideation in those who ever consumed energy drinks compared to the comparator arm (Pooled OR = 1.96, 95%CI: 1.33–2.90). Those who consumed 21 to 30 cups per month had statistically significant odds of suicidal ideation (Pooled OR = 2.78, 95%CI: 1.13–6.81), 11 to 20 cups per month (Pooled OR = 2.06, 95%CI: 1.13–3.78) and 1 to 10 cups per month (Pooled OR = 1.37, 95%CI: 1.09–1.71). Meta-regression showed a significant association between a higher amount consumed per month and higher suicidal ideation (Appendix A).

### 3.3. Coffee Consumption and Suicide Attempts

Four studies [17,18,24,26] were included to evaluate the RR of suicide attempts after coffee consumption. Meta-analyses of the four studies [17,18,24,26] (Figure 4) demonstrated a statistically significant decrease in the risk of suicide in those who ever consumed coffee compared to the comparator arm (Pooled RR = 0.72, 95%CI: 0.53–0.98). Subgroup meta-analyses by the amount consumed per month suggested that those who consumed 61 to 90 cups per month (Pooled RR = 0.51, 95%CI: 0.39–0.66) or 91 to 120 cups per month (Pooled RR = 0.57, 95%CI: 0.37–0.86) had statistically significant lower risk of suicide attempts. Those who consumed between 1 to 10 cups, 11 to 20 cups or 21 to 30 cups did not reach statistical significance. Meta-regression showed a significant association between higher amounts consumed per month and lower suicide attempts (Appendix A).

### 3.4. Systematic Review on the Factors Affecting Consumption

#### 3.4.1. Substance Usage

Five studies [16,24,29,35,36] evaluated the association between substance usage and caffeine consumption (Appendix A). These substances include alcohol [16,24,29,35,36], cigarettes [24,29,35,36], drugs [29] and marijuana [36]. All five studies [16,24,29,35,36] reported a significant association between higher substance usage and higher caffeine intake.

#### 3.4.2. Gender

Four studies [16,29,33,35] evaluated the association between gender and caffeine consumption (Appendix A). All four studies [16,29,33,35] found that males were significantly more likely to consume higher amounts of caffeinated products as compared to females.

#### 3.4.3. Risk-of-Bias

The quality of the 17 studies [16,17,18,24,25,26,27,28,29,30,31,32,33,34,35,36,37] was assessed using the JBI checklist and presented in Appendix A. Overall, there was no significant risk of bias identified. The certainty of evidence using the GRADE framework was low for the association between both coffee and energy drink consumption with suicide and very low for the association between energy drink consumption and suicide ideation (Appendix A). These are expected results due to the study designs, suggesting a higher propensity for future studies to alter the conclusions.

## 4. Discussion

This systematic review and meta-analysis sought to investigate the risk of suicidality with the consumption of caffeinated drinks, such as coffee and energy drinks. The results demonstrated that coffee consumption decreases the risk of suicide attempts. Interestingly, stratification by amount consumed per month revealed that this phenomenon was only relevant in individuals who consumed more than 60 cups of coffee per month. In contrast, consuming energy drinks was associated with an increased risk of both suicide ideation and attempts, starting from as low as one cup per month. Systematic review highlighted that male gender and substance usage significantly increased caffeine consumption. While the authors planned to also study associations of other caffeine-containing beverages and food items, such as tea and caffeine pills, no studies were identified from the literature search.

Coffee consumption is prevalent worldwide, with varying consumption patterns across different countries. Its postulated effects include enhanced cognitive functioning and protective effects against diseases such as Parkinson’s, Alzheimer’s, diabetes mellitus and even cancer [38]. With regards to suicidality, a meta-analysis by Wang et al. found that for individuals who consumed coffee, every increment in cup/day intake was associated with a 8% decrease in risk of depression [39]. The authors’ study produced similar results, with meta-analysis demonstrating a significant relationship between coffee consumption and a decreased risk of suicide attempts. The neuropharmacological effects of caffeine in enhancing neurotransmitter activity [40], have been shown to improve mood and limit depressive symptoms, increasing alertness and overall well-being [41]. Caffeine stimulates the central nervous system by antagonizing adenosine A1 and A2A receptors, reducing inhibitory signaling and increasing the release of excitatory neurotransmitters like dopamine and glutamate [42,43]. This accounts for the increased alertness, improved concentration, elevated mood, and reduced perception of fatigue experienced after caffeine consumption. Interestingly, stratification by amount consumed revealed that this association was only significant in individuals who consumed more than 60 cups of coffee per month. Previous research has shown that the relationship between coffee and various health outcomes is non-linear, such as a study by Chen et al., which found the association between separate coffee consumption and the risk of mortality to be J-shaped [42]. The authors’ results suggest the presence of a threshold effect, where moderate levels of coffee intake might not confer the same protective effects against suicidality as high-level consumption.

Conversely, the analysis revealed a concerning correlation between energy drink consumption and increased risks of both suicidal ideation and attempts. This was significant even for minimal consumption at 1 to 10 cups per month. Furthermore, the findings suggest a linear relationship between the relative risk of suicidality and the consumption of energy drinks. This means that the higher the levels of energy drink consumption, the higher the risk of suicidality. Energy drinks are designed for performance enhancement and may contain other stimulants on top of caffeine [43], which have been reported to exacerbate anxiety and agitation in their consumers [44]. On top of this, energy drinks typically contain large amounts of sugar [4], which has also been postulated to cause negative emotions such as fear, stress and anxiety [45]. Several studies have also drawn significant associations between energy drink consumption and mental health problems among adolescents [46]. The negative effect on mood and functioning could be a possible explanation for this observed correlation, increasing the risk of suicidality. With the rising popularity of energy drinks, particularly in the younger population [4], the implications of our findings are particularly concerning. These findings underscore the need for public health initiatives to raise awareness about the negative health effects of consuming energy drinks, particularly among vulnerable populations such as adolescents. In future studies and analyses, stratification by demographic or socio-economic factors such as age could be highlighted to identify specific factors that can directly affect suicidality outcomes.

Concerning demographic factors, the analysis also identified a noteworthy correlation between gender and caffeine consumption. Among all the participants who consumed energy drinks, males were likelier to exhibit suicidality than females. Current literature suggests the male psyche is more prone to suicidality [47], with studies indicating that men die by suicide at rates approximately 3.5 times higher than women [48]. This gender disparity is often linked to societal pressures associated with traditional masculine roles, promoting maladaptive coping mechanisms such as emotional suppression or a reluctance to seek help [49]. Additionally, males may be more susceptible to the effects of added legal stimulants and caffeine in energy drinks due to inherent differences in physiology and metabolism [50]. With males being more likely to exhibit risk-taking and impulsive behaviour [51], consumption of caffeine could elevate their risk of suicidality. Further studies are warranted to explore this association further, as understanding the interplay between gender and suicidality is imperative to the development of targeted interventions aimed at reducing suicidality.

Furthermore, this study identified an association between caffeine consumption and substance use behaviours. Individuals who consumed these beverages were more likely to use substances, specifically consuming alcohol or smoking cigarettes. This association raises concerns for comorbidities that could exacerbate the risk of suicidality in individuals. On the other hand, it also warrants careful consideration of confounders that could affect suicidality rates. Alcohol use has also been associated with a higher risk of major depression [52], which could, in turn, increase the risk of suicidality [53]. Similarly, multiple studies by various authors, such as Lee et al. [54] and Hughes [55], have linked cigarette use to suicidality, with smokers being more likely to commit suicide than non-smokers in multiple populations. This association underscores a need for better understanding of the interplay between these factors in order to facilitate the development of targeted solutions that could help reduce suicidality in this specific subgroup of individuals.

### Limitations

The study should be interpreted in view of its limitations. Firstly, there was a lack of studies available that explored the relationship between caffeine intake and suicidality. There were insufficient studies that investigated suicidality in other sources of caffeine, such as tea or edibles. Data on self-harm outcomes were also not available for analysis. This is pertinent as prior self-harm has been shown to significantly elevate the risk of suicide attempts [56]. Targeted interventions could be devised for early detection and increased support for these individuals before progression to suicide. Future research should be performed to elucidate the burden of self-harm among caffeine consumers. Secondly, due to the lack of studies, we could not perform subgroup meta-analyses for variables that we had initially planned for. Nonetheless, we were still able to systematically review some factors. Thirdly, there was high heterogeneity in the results, as most studies reported the number of cups of caffeine consumed but did not report the specific dose. The amount of caffeine in the products may vary based on factors such as purity, brand and region produced. Future studies should consider measuring and reporting specific dosages consumed. Lastly, it is important to acknowledge that the observed associations should be interpreted with prudence as they are likely to reflect an overall lifestyle rather than a causative effect of the beverages. This is further supported by the differing patterns observed between coffee and energy drink intake, suggesting that caffeine alone is insufficient to explain the relationship.

## 5. Conclusions

This review studied the associations between coffee and energy drink intake with suicide risk and suicidal ideation. The results highlighted that coffee consumption of more than 60 cups per month decreases the risk of suicide attempts. In contrast, consuming energy drinks was associated with an increased risk of both suicide ideation and attempts, starting from as low as one cup per month. Further studies would be essential to elucidate the psychosocial factors and causative links underlying this association. Notably, factors such as male gender and substance usage significantly increased caffeine consumption. Further research is still necessary to identify more prognostic factors of caffeine consumption on suicidality. Understanding the relationship between caffeine consumption and mental health outcomes is crucial to developing public health strategies to boost the mental health of consumers.

## Figures and Tables

**Figure 1 nutrients-17-01911-f001:**
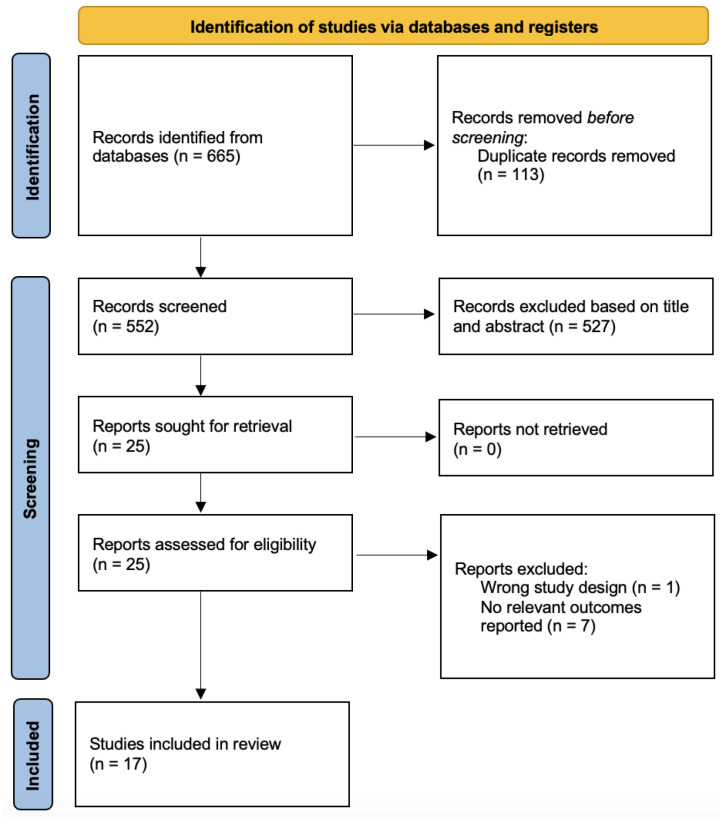
PRISMA Flowchart.

**Figure 2 nutrients-17-01911-f002:**
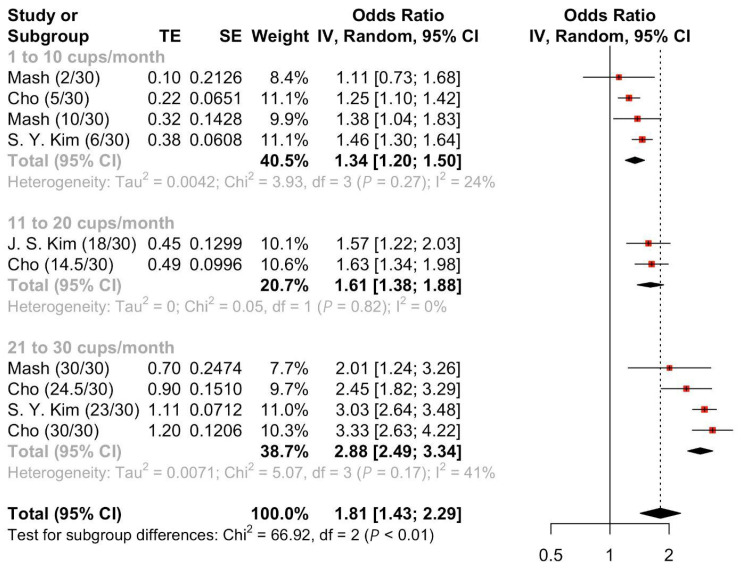
Pooled odds ratio of suicide attempts following energy drink consumption stratified by amount consumed per month.

**Figure 3 nutrients-17-01911-f003:**
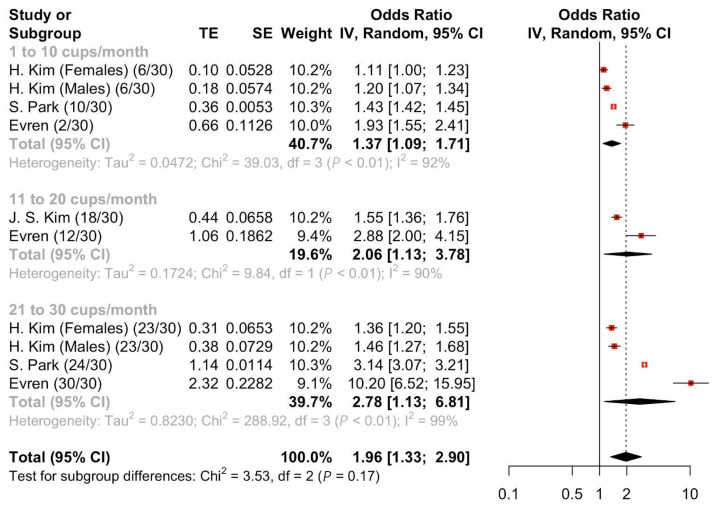
Pooled odds ratio of suicidal ideation following energy drink consumption stratified by amount consumed per month.

**Figure 4 nutrients-17-01911-f004:**
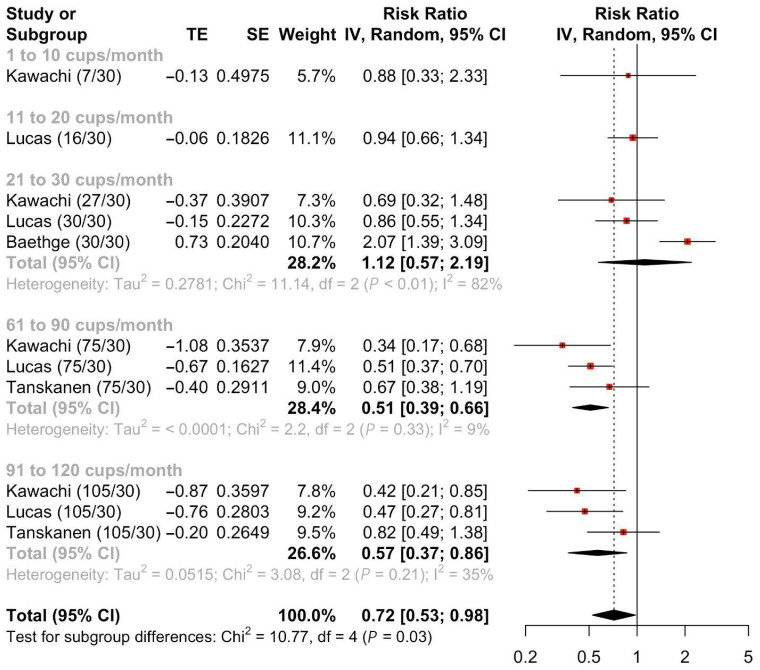
Pooled risk ratio of suicide attempts following coffee consumption stratified by amount consumed per month.

**Table 1 nutrients-17-01911-t001:** Main characteristics of the included studies.

Author	Publication Year	Region of Study	Gender (Male) %	Age at Study (Mean ± SD)	Total Number of Participants	Characteristics of Exposed Group	Characteristics of Controls	Reported Amount Consumed per Month (Mean Cups)	Outcomes Investigated
**Coffee**									
Baethge [24]	2009	Germany	49.1	44.5 ± 14.7	352	Bipolar	Non-user	30	Suicide
Kawachi [17]	1996	USA	0	NA	86,626	Women	Almost never	7, 27, 75, 105	Suicide
Lucas [18]	2014	USA	21	48.3 ± 11.7	134,604	General population	<1 cup a day	16, 30, 75, 105	Suicide
H. Park [25]	2019	Korea	58	40.2 ± 6.8	80,173	General population	<1 cup a day	30, 75, 105	suicidal ideation
Tanskanen [26]	2000	Finland	NR	NR	43,166	General population	0–1 cups a day	75, 105, 180, 240, >300	Suicide
Klatsky [27]	1993	USA	0.6	NR	128,934	General population	Non-user	30, 60, 150, >180	Suicide
**Energy Drinks**									
Cho [28]	2020	Korea	51	15 ± 0.3	62,276	Students	Non-user	5, 14.5, 24.5, 30	Suicide
Evren [29]	2015	Turkey	52.7	10th grade	4957	Students	Non-user	0, 2, 12, 30	suicidal ideation, Self-harm
H. Kim [30]	2020	Korea	49.2	NA	53,312	Adolescents	NA	6, 23	suicidal ideation
J. S. Kim [31]	2018	Korea	56.5	15 ± 1.7	8961	Adolescents	NA	18, >30	Suicide, suicidal ideation
S. Y. Kim [32]	2017	Korea	50.1	15 (NR)	121,106	Adolescents	NA	6, 23	Suicide
Masengo [33]	2020	Canada	42	15.3 ± 1.8	5538	Students	Non-user	NR	Suicide, suicidal ideation
Mash [34]	2014	USA	86	NA	508,088	Soldiers	Non-user	2, 10, 30	Suicide
S. Park [16]	2016	Korea	52	15 ± 1.7	68,043	Adolescents	Non-user	10, 27	Suicide, suicidal ideation
Kwak [35]	2021	Korea	51	15 ± 1.7	267,907	Adolescents	Non-user	NR	Suicide, suicidal ideation
**Caffeine**									
Akanni [36]	2017	Nigeria	54.8	16.9 ± 0.9	465	Students	Non-user	NR	Suicide
Beauchamp [37]	2016	USA	70	NR	40	Poison-control centre patients	NR	NR	Self-harm

Abbreviation: NA, Not available; NR, Not reported; SD, standard deviation.

## Data Availability

All data generated or analysed during this study are included in this published article [and its Appendix A].

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
