# Peer review of "Association of Coffee and Energy Drink Intake with Suicide Attempts and Suicide Ideation: A Systematic Review and Meta-Analysis"

_nutrients, 2025, doi:10.3390/nu17111911_

Round 1
Reviewer 1 Report
Comments and Suggestions for Authors
Thank you for the opportunity to review this manuscript. The authors followed the PRISMA and PROSPERO rules and conducted analysis regarding the heterogeneity, OR and RR. As it was mentioned by the authors, one limitation and I consider it very important, the author couldn't quantify the dosage of the caffeine from the coffee. I suggest the author to conduct a subgroup analysis based on the geographic region and an analysis of robustness. Based on this study I suggest the author to include in the discussion chapter a statement about these results, is there a causality or an association relationship? I suggest the authors to make a clear statement if there is any difference between caffeine and energy drinks in terms of suicidal attempts and suicidal ideation. The energy drink are quantified as cups? What is the distribution of the mean number of cups? The substance usage increased caffeine usage. I suggest the authors to include an analysis regarding the influence of the chronic mediation and caffeine usage or the influence on suicidal attempts or suicidal ideation. I suggest the author to discuss the mechanism through which caffeine influence central nervous system. See attachment.

Author Response
We thank the reviewer for the comments. Do view the attached responses document.

Reviewer 2 Report
Comments and Suggestions for Authors
Well written interesting manuscript
L65: I'm curious how they did things prospectively. It is notoriously difficult to track dietary habits. How did they categorize at risk and reference groups? That might be interesting to discuss in the Discussion section
L76: best to avoid first person. Here you put author's. That suggests one author. But in other sections you highlight "we". Again, best to avoid 1st person.
Table 1: recommend putting reference numbers next to each authors name. Ex: Smith (22)
L103: we
L122: spell out number at start of sentence
Avoid 1 sentence paragraphs: L135, 151, 163, 191
L202: we
L228: I think it is important to highlight the potential for other explanations. In the analyzed studies did they look at other factors? General health of people? Mental health? Could this be a situation where those who are at risk for suicide b/c of mental health also gravitate towards the use of energy drinks?
L236-240: same consideration. Are there other factors at play? I think this is a potentially important discussion topic. Did prior studies use multivariate regression?
L278-279: we
Author Response
We thank the reviewer for the comments. Do kindly view the attached document for the responses.
